# Non-Contact, Non-Destructive Testing in Various Industrial Sectors with Terahertz Technology

**DOI:** 10.3390/s20030712

**Published:** 2020-01-28

**Authors:** Yu Heng Tao, Anthony J. Fitzgerald, Vincent P. Wallace

**Affiliations:** Department of Physics, The University of Western Australia, Perth 6009, Australia; anthony.fitzgerald@uwa.edu.au (A.J.F.); vincent.wallace@uwa.edu.au (V.P.W.)

**Keywords:** terahertz (THz), non-destructive testing (NDT), non-contact, THz-TDS, THz CW, THz cameras, THz pulsed imaging (TPI), industrial applications, building and construction, oil & gas, manufacturing, concrete, thermal barrier coatings, electrical cables, natural gas pipelines, polymer composites, steel fabrication, multimode laser diodes, electronically controlled optical sampling, beam steering

## Abstract

In this article, we survey various non-contact, non-destructive testing methods by way of terahertz (THz) spectroscopy and imaging designed for use in various industrial sectors. A brief overview of the working principles of THz spectroscopy and imaging is provided, followed by a survey of selected applications from three industries—the building and construction industry, the energy and power industry, and the manufacturing industry. Material characterization, thickness measurement, and defect/corrosion assessment are demonstrated through the examples presented. The article concludes with a discussion of novel spectroscopy and imaging devices and techniques that are expected to accelerate industry adoption of THz systems.

## 1. Introduction

Terahertz (THz) refers to electromagnetic radiation of frequencies around 10^12^ Hz. The general definition covers the frequency range between 0.1 THz to 10 THz [1], which correspond to wavelengths between 3 mm to 0.03 mm, respectively. As the bandwidth lies between microwaves, which are generated electronically, and the infrared, generated photonically, producing and detecting THz radiation had been difficult due to a lack of suitable materials and methods, although the potential utility of this band was suggested as early as 1970 [2]. However, since the successful implementation of ultrafast lasers [2] research in the field took off in the 1990s and has been growing ever since. Today a wide range of THz systems and applications have been developed across a broad range of sectors, including biological [3], medical [4], and industry [5,6].

One of the most distinctive features of THz radiation is its ability to penetrate a wide variety of non-conductive materials. In this respect, it is similar to microwaves but the spatial resolution is much higher given the shorter wavelength [7]. In the industrial context, this allows examination and detection of defects in small devices such as integrated circuits [8], as well as structural defects such as cracks in concrete [9].

Another distinct feature of THz radiation is its strong sensitivity to water and other polar liquids [3,10]. This sensitivity stems from the fact that hydrogen bonds between water molecules resonate strongly at THz frequencies, absorbing most of the incident THz radiation in the process of dipole re-orientation. This allows for the precise measurement of moisture content and the monitoring of water ingress in a wide range of building materials and products [11].

As to the safety of THz radiation, it is non-ionizing due to its low photon energy (~4.1 meV at 1 THz [12]); this allows THz systems to be deployed in the field without the complication of stringent safety requirements as that of, for example, X-rays [13].

This article provides a brief survey of the use of THz technology in the industrial sector. The first section provides an overview of the underlying operating principles of THz systems. The second section provides recent examples of industrial applications from three industries, namely the building and construction industry, the energy and power industry, and the manufacturing industry. The third section describes emerging equipment and techniques that have the potential to improve significantly and solve the challenges leading to the widespread adoption of THz in the industrial setting.

## 2. THz Spectroscopy and Imaging

THz systems can be classified into two types—broadband systems, commonly referred to as time-domain spectroscopy (TDS); and narrowband systems, commonly referred to as continuous wave systems (CW) [5].

TDS systems probe the sample with a sub-picosecond pulse of electromagnetic radiation, which is broadband in nature, and measuring the response in the time domain yields the phase and amplitude change caused by the sample. This allows the determination of properties such as the refractive index, absorption and dielectric constant, as well as spectroscopic analysis of chemical compositions and thickness and depth information of layered structures. Tomography and topography of visibly opaque objects are also possible using time-of-flight analysis [14]. For this reason, TDS systems are the preferred type for their versatility where cost and physical footprint allow.

On the other hand, CW systems probe the sample with continuous waves of THz radiation of a single frequency or small bandwidth, generally just a change in signal amplitude is measured but it is possible to obtain phase information using a homodyne detection scheme [15]. Depending on the sources and set-up, the spectral resolution can be < 50 MHz and the output power can be higher than 10 mW. Generation of CW THz waves can be done electronically, via frequency multiplication, or optically, via photo-mixing. Without the complicated and expensive setup for pulse lasers, CW systems are generally smaller in size and lower in cost.

### 2.1. THz-TDS

Generally, TDS can be in reflection or transmission mode (or both). In each mode, the sample’s refractive and absorption coefficient index can be calculated from the time delay and attenuation of individual frequency components. Coherent detection is required to overcome thermal noise, which is more than 25.7 meV at room temperature (thermal energy = *k_B_T*) [16].

The critical components of a THz-TDS system include a THz emitter and a THz detector, an ultrafast (femtosecond) laser source, a beam splitter, a delay line, and a set of collimating and focusing lenses and mirrors. The emitter is commonly a photoconductive antenna (PCA) consisting of a micro-antenna fabricated on a semiconductor substrate with a short carrier lifetime (e.g., GaAs). The detector is typically another PCA or a non-linear crystal such as ZnTe that favours optical polarization in one direction. Any femtosecond laser can be used providing the wavelength and hence the photon energy is above the bandgap of the semiconductor substrate; typically for low temperature grown GaAs, this is titanium:sapphire (Ti:S) or frequency-doubled erbium fibre lasers ( ~800 nm). 

During operation, the laser emits a short pulse ~100 fs. This is divided by a beam splitter into a ‘pump’ beam and a ‘probe’ beam. The former photo-excites the THz emitter, which generates a pulse of THz radiation. The THz pulse is collimated and focused onto the sample. The reflected or the transmitted signal is collected and focused onto the THz detector. The THz detector is activated upon photo-excitation by the probe beam and converts the impinging THz pulse into a proportional current signal to be recorded by a computer. Sampling by the probe beam can be achieved using an optical time delay line which is typically performed by mechanically varying the length of the probe beam path relative to the pump beam; alternative methods are described later [17].

A critical development in making THz TDS systems more practical is the move away from free-space optics, where the femtosecond laser beam is guided and focused onto the emitter/detector using mirrors and lenses, to fibre optic coupled devices. This allowed for more flexibility in industrial settings. The main challenge is maintaining the short pulses and sufficient power through the fibre optic. Initially, fibre optics were used to channel short pulses of light for applications in two-photon fluorescence [18] and this was commercially developed for fs Ti:S lasers by Coherent Inc. in 1998 [19]. This method started to be adopted in the THz field driven by the need to scan objects remotely from the core unit [20,21] and all fibre compact systems appeared around 2008 [22].

### 2.2. THz-CW

CW terahertz can be generated using optical means, a combination of optics and electronics, or using solid-state devices.

Parametric conversion is an optical technique of generating CW THz waves. It involves a non-linear optical crystal such as *LiNbO_3_* being pumped by a nanosecond pulsed laser, producing polariton scattering [23]. Details of this method can be found in [24]. THz waves produced by this method have large tunability between 0.7 and 3 THz. The size of the equipment is compact enough to fit on a tabletop.

Photomixing is a combination of optics and electronics techniques to generate CW THz waves. In photomixing two semiconductor lasers of different frequencies are phase-locked and combined in a photomixer, which outputs a wave equal to the difference of the two laser frequencies. Tuning is achieved by varying one of the laser frequencies. A comprehensive review of various photomixing schemes can be found in [25].

Frequency multipliers are an electronic technique for obtaining CW THz waves. THz frequencies using this technique is realized by multiplying the input frequency through a multiplier circuit; more details can be found in [26].

Quantum cascade lasers (QCLs) are another example of generating CW THz waves by amplifying photon emission through a semiconductor heterostructure (superlattice) which was first proposed in 1971 in [27], but it took more than 23 years before the first QCL was implemented [28]. The first THz QCL was demonstrated in [29] which operated at 4.4 THz with 2 mW output power, operating at a temperature of 50 K. 

## 3. Applications in Industry

The ability of THz to penetrate most non-metallic materials allows non-contact examination of materials that are opaque in the visible range such as concrete, insulating foam, and paint. The properties of interest across the industries may be broadly categorized into three areas—layer thickness, defects and contamination, and material characterization.

Although the key parameters of interest are application-specific, the advantage of terahertz over other mature technologies in non-destructive testing (NDT) is in providing new information. For example, in determining paint layer thickness in the car manufacturing industry, existing ultrasound techniques require physical contact between the sensing head via a gel medium; this restricts the use of ultrasound on dry paint. Terahertz, on the other hand, can be used without any physical contact, providing information on fresh paint layers and the drying process [30].

Across applications, the bandwidth and the signal-to-noise (SNR) ratio are perhaps the most common parameters used for comparing different terahertz systems, but specific requirements are application dependent. 

Thickness measurements have been demonstrated with both TDS and CW systems. In a TDS setup, reflections from the layer interfaces are identified as peaks in the time domain. The thickness can be determined by the time delay as
*2d* = *∆t(c/n)*,(1)
where *d* is the layer thickness, *∆t* is the time between the reflections, and *n* is the refractive index of the material. For sub-micron layers where the reflection peaks overlap, several numerical techniques have been demonstrated [31,32].

Thickness measurements with CW offer a compact and cost-effective alternative to TDS. For phase extraction, coherent detection can be achieved via a homodyne detection scheme. As demonstrated in [33], a photomixer involving two distributed feedback (DFB) lasers achieved a resolution of 1 GHz. The same group also demonstrated a non-frequency sweep method via Gouy phase shift interferometry for fast data acquisition [34].

Fibre-coupled systems eliminate the need for bulky free-space optics and mechanical movements, yielding compact and robust NDT tools. An ultrafast, fully fibre-coupled CW THz spectrometer was demonstrated in [35]; spectra over a bandwidth of 2 THz can be acquired at a speed of 24 Hz.

The main challenge in detecting structural defects lies in generating sufficient contrast of the defects (e.g., a subsurface void) against the background. For example, cracks and voids in concrete are more apparent in a THz image when filled by water or contaminant other than air. This is due to higher absorption by the ingress material reflecting less radiation compared to that of the surrounding concrete. In the case of porous materials such as foam, the difference between defective voids and internal air pockets can be difficult to distinguish, particularly if the air pockets are of a submillimeter size or larger also scatter the THz beam. Examples for this are given in the “Insulating Foam” section below.

Much work on material characterization with THz has aimed to establish a database, of which other applications can make use. One example is in production quality control, where a sample’s electrical properties such as conductivity must be investigated to meet specific criteria before proceeding to the next stage of production. The following sections provide examples of applications grouped by industry.

### 3.1. Building and Construction Industry

#### 3.1.1. Concrete

Concrete is one of the most ancient materials used in the construction industry. Evidence of its use in housing structures dates back to 6500 BC in the region of Syria and Jordan [36]. Due to its strength and durability, concrete is still one of the most widely used materials in modern infrastructures. Ensuring the structural integrity of aging concrete is of great importance to longevity and public safety. 

Defects in concrete such as cracks may develop over time due to corroding steel rods as well as various environmental factors. Conventional techniques for detecting internal defects include ultrasonic-echo, infrared thermography, and ground-penetrating radar (GPR). However, these techniques either lack penetration depth/resolution, are cumbersome to set up or require complex data processing. In [37], it was demonstrated that THz can detect internal defects in concrete up to 100 mm thick. CW in transmission mode were used to scan and generate 2D images of concrete samples with varying thickness.

A common failure mode for concrete begins during manufacturing at the hydration stage, when mixing the aggregate with water and cement. Insufficient water can reduce the strength and durability of the concrete and increases the likelihood of water ingress that can lead to rapid deterioration and corrosion if the concrete contains reinforcing steel rods. The conventional method of inspection for water content is a high-frequency capacitive measurement [38], which requires direct contact with the sample as well as prior knowledge of the sample characteristics (e.g., thickness, specific gravity, etc.). 

In [39], the authors determined the water content in a concrete sample using CW terahertz in transmission mode. It was found that the absorption coefficient increases with water content up to 10% by mass, after which absorption sharply increases. This sharp increase was attributed to free water absorption. The authors also demonstrated that the reflection intensity decreases as the water evaporates from the concrete sample. This was attributed to the incident wave having to travel further into the concrete to reach the concrete–water interface, thus increasing scattering and absorption by the concrete. This highlights a potential issue with interpreting the reflected intensity. For example, a higher reflection can either be caused by some moisture near the surface or a deeper saturated layer. 

This issue could be resolved by adopting a time-domain approach. Specifically, the location of a concrete–water interface may be calculated from the time delay alone, while the depth-calibrated reflection intensity reveals the water content at each depth.

Due to the porous nature of concrete, steel rod reinforced concrete is susceptible to corrosion, especially in coastal regions, due to the ingress of chloride ions [40]. X-ray fluorescence and infrared thermography are two non-destructive methods of monitoring chloride ions concentration in concrete. However, these methods are dangerous to operate and lack penetration depth, respectively. In [40], it was demonstrated that chloride ions content can be monitored using THz TDS. The absorption coefficient increases linearly with the chloride ion concentration over the frequency range of 0.05–0.2 THz. The authors were able to achieve a 2–5 mm penetration depth with a THz TDS system. Conceptually, an imaging set up could generate maps of contrast between areas of high and low reflectivity to locate and monitor chloride ions ingress. For the same reason as with the water ingress mentioned above, a time-domain approach would be preferable for determining the concentration and depth.

In [41], the authors demonstrated the imaging of corroded steel rods buried in concrete using a CW system in reflection mode. The corroded section of the steel rods appears as an area of low reflection in the image (see Figure 1). The intensity of the reflection is correlated to the degree of corrosion. The setup utilizes a 0.2 THz-band GaAs TUNNETT diode oscillator at room temperature.

#### 3.1.2. Insulation Foam

Foam material, such as expanded polyurethane, is widely used for building and industrial insulation due to it being lightweight and its high thermal and acoustic absorbing capacity. To detect internal defects, thermal imaging is ineffective due to the slow thermal diffusion of foam. The porous nature of foam makes it completely transparent to X-rays but opaque to visible light due to strong scattering by micro-particles. Terahertz radiation has shown good potential for this purpose, as shown in Figure 2. Figure 2 shows that the expanded polyurethane foam has a refractive very close to that of air; the attenuation coefficient increases with frequency, which is predominantly due to scattering. Radiation up to 0.5 THz can penetrate the foam tens of centimeters deep with minimal attenuation; defects within this depth can be highlighted, if scattering noise can be filtered out.

Detecting defects in foam material started gaining focus in the THz research community at the time of the NASA Columbia Space Shuttle Disaster in 2003, where the shuttle’s wing was damaged by a piece of foam insulation that fell off the shuttle’s main fuel tank during launch [42]. NDT THz imaging was subsequently incorporated into the standard pre-flight test as part of the testing of the shuttle insulation [21]. In [43], THz equipment and technique used by NASA to perform NDT on critical foam pieces are described. The setup utilizes a transceiver head connected to a main control unit via an umbilical cord (see Figure 3a). The transceiver head is used to scan the area of interest. Test panels with foam voids was imaged in [21] using this setup. The vertical black line through the middle is a flange joining two pieces of test panels. The two vertical black lines on either side of it are milled trenches in the foam, which are covered with glue. Voids can be readily identified as dark streaks running parallel to the six stringers (see Figure 3b). 

Numerous effective medium models have been explored to model the scattering in foam. In [44,45], Rayleigh scattering was found to accurately predict magnitude loss at low terahertz frequencies with pore sizes ranging from 1 µm to 15 µm using THz-TDS transmission, and high frequencies can be predicted reasonably well by Torquato–Kreibig–Fresnel (TKF) model. In [45], it was confirmed that the larger the micro-particles and/or the higher the frequency, the more attenuated the signal, which agrees with the Mie scattering model. In [46], knit lines and voids in polyurethane films can be detected using a model based on the Clausius–Mossotti equation.

Various types of polymer foam used for building insulation have also been characterized, for example in [47] which shows that fibrous, cellular, and granular types can be distinguished. The optical parameters are shown to be closely related to the insulation performance, indicating THz as a useful tool in assessing the performance of newly developed foam materials and structures. 

#### 3.1.3. Plastic and Adhesives

Due to the strength and resistance to corrosion plastic joints have become increasingly used on critical pipes such as gas and water pipes. In plastic welding, contamination or imperfection in the welding joints can dramatically reduce the structural integrity of the weld. Quality inspection on such joints can prevent significant hazardous incidents. Ultrasound is a standard method of NDT but it is strongly limited in resolution and it has a high attenuation coefficient in thermoplastic. Wieztke et al. [48] demonstrated the successful detection of contamination and delamination between layers of high-density polyethylene using THz. In their experiments metal staples, sand, and air gap of ~350 µm between the HDPE layers can be readily identified using THz-TDS transmission (see Figure 4).

### 3.2. Energy and Power Industry

#### 3.2.1. Thermal Barrier Coatings

In thermal power generation, gas turbine blades operate in high temperatures exceeding 1100 °C as part of the combustion process. To protect the blade metal from oxidation, a thermal barrier coating (TBC) is applied to the blades, which typically consists of a ceramic layer-topcoat, an alloy layer-bond coat, and a bulk metal substrate. In [49], the authors demonstrated the use of THz pulse reflection to measure the topcoat thickness; the results were verified with microscopic imaging to be within measurement error. In [50], a similar measurement was carried out with a custom erosion platform to simulate topcoat erosion in the real operating environment. The measurement was validated with the micrometer (see Figure 5 and Table 1).

As the topcoat is made porous to accommodate thermal expansion and contraction, alumina oxide forms at the interface between the topcoat and bond coat. This oxide creates internal stress between the layers which is exacerbated by the thermal cycles, which can eventually lead to voids and delamination. In [51], the authors demonstrated the ability to use reflectometry to monitor the formation of alumina oxide and air gap of a few microns in TBC samples. The corrosion was mimicked in a furnace, and the results were verified with images taken using a scanning electron microscope (SEM).

#### 3.2.2. Electrical Cables

Detection of cable insulation failure has become increasingly important as electrical distribution networks age. The current method of visual inspection is restricted to areas where insulation damage is apparent and may require physical handling of the cables. Therefore non-destructive, non-contact testing for cable integrity is much needed. THz radiation is ideally suited for the examination of the cable internal because it can penetrate cable insulation, commonly made of polymers such as polyethylene. The copper conductors are highly reflective of THz radiation, but as they corrode, the oxide formed on the wires strongly absorbs THz radiation to generate contrast [52].

In [41], THz reflection images of deliberately damaged copper cable and corroded copper cables are shown (see Figure 6). Gaps larger than 1.5 mm in individual strands were visible. The degree of corrosion also corresponds to the intensity of the THz reflectivity, as demonstrated in [53]. The images were generated using a 0.14 THz IMPATT oscillator and a Schottky barrier detector. 

#### 3.2.3. Natural Gas Pipelines

In natural gas pipelines, water vapor can induce the formation of ice particles, reducing or even blocking gas flow and it can also accelerate corrosion on the internal pipe surface. Pipeline operators should need to monitor the water vapor in order to ensure it remains below a pre-determined threshold.

Conventional methods of monitoring water vapor include the use of chilled mirrors, impedance hygrometers, and IR and microwave spectroscopic hydrometers. Except for microwave hydrometers, these methods lack selectivity for water, such that other gases may interfere and produce incorrect readings. Microwave hydrometers have good selectivity for water, but their sensitivity is low.

In [54], the authors demonstrated the feasibility of using THz-TDS to monitor water vapor in a pressurized gas pipeline. A 14.7 cm long gas cell filled with methane was tested with a commercial TDS transmission system. The transmitted spectrum was recorded with varying humidity and pressures simulating industrial operating conditions. A novel data processing method was proposed to account for the continuum absorption of water vapor at high pressure. It was found that at an industrial operating pressure of 100 bar, water vapor content as low as 62 ppm in the gas cell can be estimated accurately. Furthermore, the relative humidity can be determined from the intensity of the absorption peaks (see Figure 7), which is increasingly sensitive as the humidity increases. 

### 3.3. Manufacturing Industry

#### 3.3.1. Steel

In steel fabrication, small protrusions or dents can occur on steel sheets as a result of varying manufacturing conditions. It is preferable to identify such imperfections as early as possible to minimize waste further along the production line; this calls for inline quality control. Due to surface roughness, the detection of small imperfections using visible light is difficult. In [55], Hasegawa et al. demonstrated the detection of a small protrusion (30 µm in height) on a sheet of steel by a method termed dark-field THz imaging, where scattered and refracted THz radiation are collected for analysis. In Figure 8, the protrusion can be seen. 

In [57], Whelan et al. demonstrated the conductivity mapping of graphene sheets on a polyethylene terephthalate (PET) film substrate using TDS transmission mode, accounting for internal reflections within the substrate. The results agree with four-point probe measurement of sheet conductivity, and thin scratches are visible in the map as lines of low conductivity.

#### 3.3.2. Polymer Composites

Polymer composites consist of a host material, usually polymer resin, and a reinforcing fibre material such as glass or carbon. The addition of fibre enhances the properties of the material such as increased strength, lighter weights, enhanced resistance to environmental conditions etc. For this reason, polymer composites are replacing metals and concrete in structures and load-bearing parts in many industry branches including construction [58], aviation [59], and marine [60].

Various factors can contribute to the degradation of composite materials; these include environmental contamination, abnormal stress loading, and non-ideal manufacturing conditions.

An example of non-ideal manufacturing conditions occurs during the mold-injection process, where an inhomogeneous distribution of the fibers and their alignment, which can be a source of weakness leading to failure of the joint. An NDT method to examine for this anisotropy within optically opaque composites is needed, particularly for critical parts. Katletz et al. demonstrated the use of a circularly polarized THz pulse reflection set up to map the glass fibre optical alignment within an HDPE sample [61]. This work has been furthered by measurements done in [62], where black rubber with carbon filler was stretched to produce birefringence, which can be observed clearly on images produced (see Figure 9). As black rubber with black conductive fibres is commonly used in tires and birefringence are readily induced by mechanically stretching the material, birefringence mapping by THz provides a potential NDT method to estimate the internal strain the sample has undergone.

In regards to the detection of contamination, Mieloszyk et al. [63] demonstrated the imaging of a water drop placed within a stack of glass fibre reinforced polymer (GFRP). Detecting water ingress into GFRP predicts microcracking induced by the water absorption and desorption cycle. In the experiment, a drop of water was placed between two layers during the production process, and the shape of the water drop can be observed using THz TDS reflection imaging, in both rough and smooth surfaces of the top layer.

Novel signal processing techniques have also been explored that can reduce image acquisition time significantly; e.g., compressive sampling (CS), as demonstrated in [64]. The authors compared the performance of various CS solvers in reconstructing simulated images with different shape cracks. The evaluation considered the number of measurements required, errors in the reconstructed image, and computational resources. A reduction of >70% in measurement time compared to raster-scanning is typical with acceptable computational resource consumption.

#### 3.3.3. Automotive Paint

Terahertz has also been shown to be promising for in-line quality control in the automotive industry [65]. On automotive production lines, the paint layer thickness is measured right after the painting of each layer. This is to ensure the right amount of each paint type is applied evenly across the vehicle body. Conventional methods relying on eddy-current, magnetic gauges, or ultrasonic techniques are challenging to integrate into the multi-step painting process.

Krimi et al. demonstrated the successful point measurement of a multilayer paint coating with a fibre-coupled spectrometer [30]. The TDS spectrometer was set up in reflection mode. A measurement time of 1s and a minimum thickness of 4 µm was achieved. More importantly, the proposed self-calibration method takes into account the merging effect between wet layers, allowing accurate measurement in practical situations where the wet-on-wet spray is common. Coatings on metallic and dielectric substrates were shown to be in good agreement with the results obtained from the analysis of micrographic images. The thickness range was also impressive; a dielectric substrate of 3 mm with a thin coating of 10 µm was successfully measured; this area of application is described in more detail in [6].

## 4. Outlook

The utility of terahertz radiation in NDT has been demonstrated, as noted in the preceding sections. However, several obstacles prevent wide industry adoption.

Firstly, the relatively expensive equipment deters businesses, which are commonly driven by immediate financial benefits. While various prototypes demonstrated good precision or resolution. Trading precision for cost is a strategy that more and more manufacturers are pursuing. Also, being able to utilize existing manufacturing equipment and processes is expected to lower the barrier for manufacturing. An example of this is the FET-based THz camera discussed below.

Secondly, the speed of measurement is critical in an industrial setting that may be less obvious in a research environment. The labour cost involved is magnified by the usage of the system. The net financial benefit of each measurement would need to be positive and large enough for the timely recoupment of the investment.

Finally, the portability and the robustness of the system typically falls short given the environments in which systems will be potentially used. The equipment would be frequently exposed to mechanical and acoustic vibration, heat, humidity, and the weather. Measurements may need to be done in hard-to-get-to places with limited space. Being portable and robust encourages its use, while lowering the training requirement for the operational and maintenance personnel.

The sections that follow present examples of recent advancement in terahertz technology that addresses some of the obstacles described above.

### 4.1. Multimode Laser Diodes

The average price of a basic THz system is around US$75k [6], it is still high compared to other established NDT techniques such as X-rays and ultrasound, which are a fraction of the cost [66]. The bulk of the system cost is for the ultrafast lasers used for THz generation. Such pricing, along with the lack of portability makes THz systems cost-prohibitive for many industries, and it is one of the main reasons why the transfer from laboratory to industry has been slow.

To lower cost, multimode laser diodes have been proposed as an alternative laser source as ultra-fast (femtosecond) lasers constitute about half of the system cost [67]. Multimode laser diodes are relatively cheap and commercially available. TDS based on multimode lasers is also an active area of research [68,69]. Signals generated by multimode laser diodes are not pulses but similar, hence these systems are known as Quasi TDS (QTDS); a more theoretical background description of QTDS can be found in [70]. An example of QTDS implementation was demonstrated in [69], where a compact system was built with widely available equipment including Arduino DUE board in place of a lock-in amplifier. The size can be further reduced to that of a PCB with a Raspberry Pi in place of a personal computer.

### 4.2. Electronically Controlled Optical Sampling (ECOPS) and Asynchronous Optical Sampling (ASOPS)

The optical delay line is one of the most critical components in a THz TDS system, and the quality thereof directly affects the accuracy and speed of the systems. Most optical delay lines use a mechanical component that linearly alters the length of the probe beam path. High costs are dedicated to the speed, precision, and consistency of the mechanical movement.

Non-mechanical delay lines, such as electronically controlled optical sampling (ECOPS) and asynchronous optical sampling (ASOPS), have been developed as a better alternative in terms of measurement speed and robustness. In both ECOPS and ASOPS two femtosecond lasers are used one to the emitter the other for the detector; and the repetition rate of the two lasers can be varied. This achieves sampling like a mechanical delay line albeit at a much faster rate.;the typical measurement rate is in the kHz range. For example, the latest ASOPS spectrometer TERA ASOPS from MenloSystems has a bandwidth of >3 THz, detection time window up to 10 ns, and 60 dB dynamic range at >600 Hz scanning rate [71]. The speed of ECOPS was demonstrated in [66]. The authors compared two TOPTICA TDS systems, one with a conventional delay line and one ECOPS-based, in measuring the thickness of silicon wafers. The ECOPS-based system was 160 times faster albeit with slightly less accuracy; this suggests ECOPS particular suitability for inline quality control.

### 4.3. THz Cameras

In imaging applications, focal plane arrays (FPA) provide a much faster alternative to raster scanning that allows real-time imaging over a larger area and the studying of dynamics. An example was demonstrated in [72], where the hydration of Nafion membrane used in fuel cells were recorded in real-time with a compact THz camera.

The earliest THz imaging using a CCD camera was based on 2D electro-optic sampling, with ZnTe as the EO crystal and the camera at the focal plane. The source of illumination can broadband pulse [73,74,75] or narrowband CW [76]. The THz beam causes a spatially varying refractive index on the EO crystal (aka the Pockels effect), which is captured by a CCD camera.

In [77] THz imaging with microbolometer arrays was demonstrated; the array elements were IR microbolometers made of vanadium oxide. For illumination, a CO_2_ laser was used, although QCL has also been demonstrated in [78]. Several institutions such as CEA-LETI [79], Institut Natinoal d’Optique (INO) [80], and Swiss Terahertz [81] are actively developing microbolometers-based FPAs.

FET-based cameras rely on the interaction of THz beam with plasma waves in the transistor channel; both narrowband and broadband detection have been demonstrated [7]. The transistor material ranges from III-IV semiconductors [82,83,84], INAs nanowire [83,85], graphene [86], to Si-based. Due to good compatibility with the existing CMOS process Si-based FET is expected to be the first to gain mass-market adoption.

Schottky barrier diodes (SBD) as shown in [87,88] are another common array element; in [88] the system had a 1 mm image resolution when scanning objects at 25 cm/s, indicating good potential as a quality inspection tool on a production line.

Other array elements such as pyroelectric crystals [89,90,91], microelectromechanical systems (MEMS) [92], and superconducting films [93] have also been reported. Pyroelectric crystals have limited sensitivity due to the relatively large pixel size, while MEMS have shown incredible response speed in the kHz range. Superconducting film in the form of kinetic inductance detectors (KID) provides excellent SNR; however, cryogenic cooling is required.

Terahertz imaging can be realized using non-THz cameras via frequency up-conversion, as demonstrated with an IR camera in [94] and an optical camera in [95]. IR conversion utilizes metal-insulator-metal (MIM) antennas with silver nanoparticles grating, which require proper matching of resonance between grating geometry and the THz source for good resolution. Converting THz to the visible part of the spectrum can be achieved with a plane of laser-excited atoms. In [95], caesium atoms were excited by three lasers and fluoresced green light upon incident terahertz radiation. An optical camera then captured the spatial distribution of fluorescence intensity. The kHz frame rate achieved showed promising potential in fast-moving production lines.

Stretchable detectors have also been demonstrated in [96,97,98]; they consist of electrodes sitting on a film of carbon nanotubes that produces a voltage based on photo-thermoelectric effect and they have characteristic absorption between 0.1–300 THz and no biasing requirement. The flexibility and stretchability also mean these can be attached to objects of various shapes such as industrial pipes and electrical conduits; defect or impurities beneath the detector show up under THz illumination as demonstrated in [98].

For applications where space is extremely limited, lens-less systems may be ideal as it does not contain any optical components. A Fresnel aperture combined with phase reconstruction algorithms [99] can be used to achieve comparable resolution as that of a lens system.

Novel image processing techniques should also be discussed. For example, deconvolution with a point spread function (PSF) enhances the resolution of an image as demonstrated in [8]. The authors modeled a PSF based on the Gaussian distribution of the THz beam. The result was a dramatic improvement in image resolution compared to using a recorded PSF as is done conventionally.

Super-resolution reconstruction methods can achieve a higher resolution image just by combining lower resolution images. In [100], four 64 × 64 pixel images were combined via the iterative back-projection algorithm to produce a 128 × 128 pixel image of much-improved clarity.

### 4.4. Beam Steering

In applications where spectroscopic information is required, point detectors are still preferred over THz cameras, as THz cameras provide no phase information. To reduce the image acquisition time, beam steering is a potential alternative to raster scanning.

Beam steering is based on the idea that steering an emitted THz beam over a sample area is quicker than mechanically moving the sample. This scanning method was demonstrated in [101], where an area of 288 × 207 mm was imaged in 3.13 s versus 42 min by conventional raster scanning; which was achieved using a polygonal mirror and an f-theta scanning lens. However, non-mechanical means such as distributed emitter arrays [102] and liquid crystal devices [103] have also been reported.

### 4.5. Quantum Cascade Lasers (QCL)

QCLs are a powerful, robust, and versatile source of mid-IR to far-IR radiation. In the THz range QCLs show excellent performances over 1–5 THz with the highest output power >1 W [5]. Unlike conventional semiconductor diode lasers, where the wavelength is determined by the material bandgap, the wavelength emitted in QCL can be chosen by designing the thickness and geometry of the active region hetero-structure.

In QCLs, the emission of a photon occurs entirely within the conduction band by means of inter-sub-band transitions. These transitions occur between different energy levels within quantum wells. An electron makes the transition between a pair of coupled quantum wells by tunneling. The same electron is recycled and goes through the same process repeatedly to emit more photons. Structurally, each quantum well is essentially a nanometer-thin layer of semiconductor. The layers are stacked by way of molecular beam epitaxy (MBE) [28], resulting in a series of cascaded quantum wells, hence the name.

Since their conception in 1994, QCL has attracted significant research focus. Today, manufacturers such as Lytid SAS can offer terahertz sources in the 2–5 THz range at >1 mW [104]. Recently, the development of a multi-wavelength QCL array shows promise. QCL arrays combine the versatility of QCL with the flexibility of array design to achieve a range of attributes including continuous frequency tuning, high peak output power, even power distribution, and incredible measurement speed, all on a chip [105]. However, cooling to <100 K is still required for operation in the terahertz range, which is a major drawback in an industrial environment. Operation at room-temperature would require a breakthrough in material science.

## 5. Conclusions

In this review, we provided an update on the adoption of non-contact, non-destructive testing by terahertz technology in the various industrial sectors. The advantage of Terahertz over established NDT technologies were described, namely its superior penetration depth in certain materials and spatial resolution. The key components and basic working principles of the two most common types of THz systems—broadband and narrowband, were described. This is followed by a survey of recent application examples in three industries—the building and construction industry, the energy and power industry, and the manufacturing industry. The selected examples demonstrated THz capacity in thickness determination, defect and corrosion mapping, and material characterization. Finally, trends and development of new components and devices were presented. These developments show great potential in the context of industrial applications, where cost, size, measurement speed, and robustness of the system are important.

As THz research continues to advance, smaller, more efficient, yet cheaper systems are becoming increasingly available. To increase the rate of industry adoption, OEM and the research community need to be proactive in exploring unknown demands. In the face of increasing competition, this is particularly important for small to medium system suppliers, who have a competitive edge in customizing products and solutions for niche markets.

## Figures and Tables

**Figure 1 sensors-20-00712-f001:**
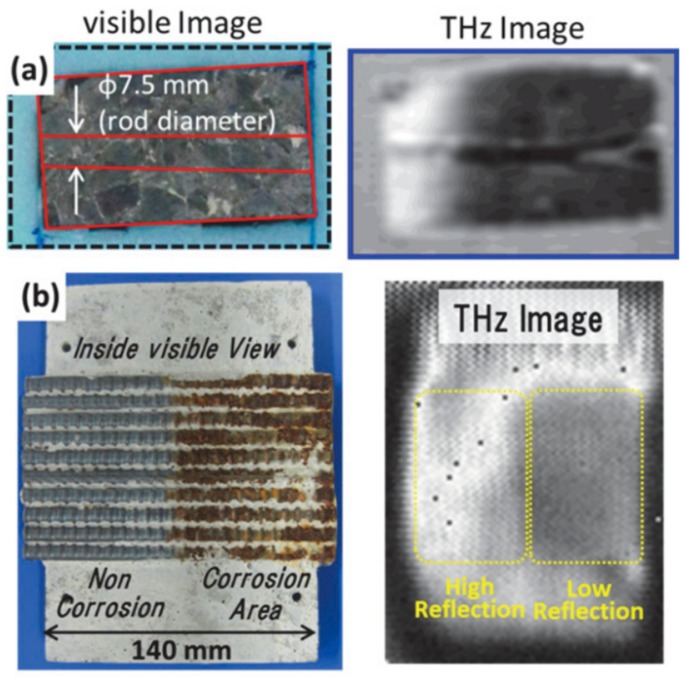
Steel rods inside concrete imaged using THz TDS reflection mode: (**a**) Visible view of the concrete containing a steel rod (left) and the THz image showing the steel rod within (right); (**b**) Cut out view of the concrete showing corroded steel rods and the non-corroded steel rods (left), and the THz image—corroded steel rods produce significantly lower reflection compared to non-corroded steel rods. Reproduced with permission from [41].

**Figure 2 sensors-20-00712-f002:**
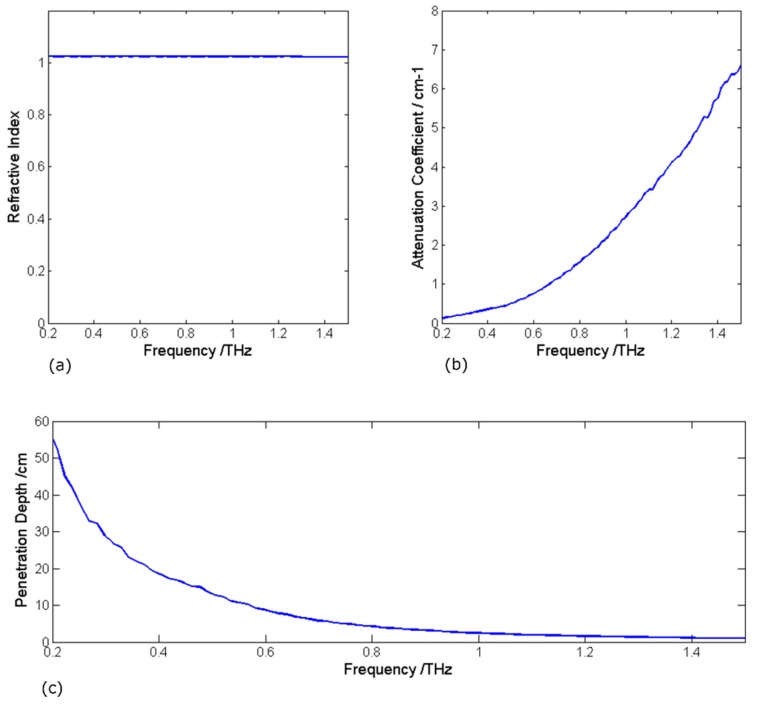
(**a**) Refractive index of insulating foam, with a constant refractive index of 1.02 up to 1.5 THz; (**b**) Attenuation coefficient, which is low compared to many materials, rising with frequency primarily due to scattering; (**c**) Presents the penetration depth for the system SNR of almost 1000:1 indicating that lower frequencies < 0.5 THz are best for penetrating the insulation of the order of tens of centimeters thick.

**Figure 3 sensors-20-00712-f003:**
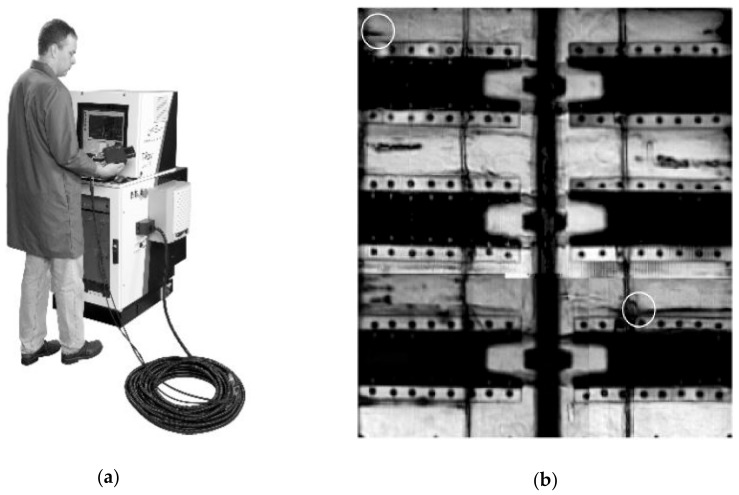
Pulsed THz imaging of space shuttle foam for defect detection: (**a**) Photograph of the equipment. The transceiver head is connected to the main control unit via an umbilical cord; (**b**) THz image of a test panel with voids in the spray foam. Voids appear as horizontal dark streaks. White circles mark the areas where the voids crack through the milled surface of the panel. Reproduced with permission from [21].

**Figure 4 sensors-20-00712-f004:**
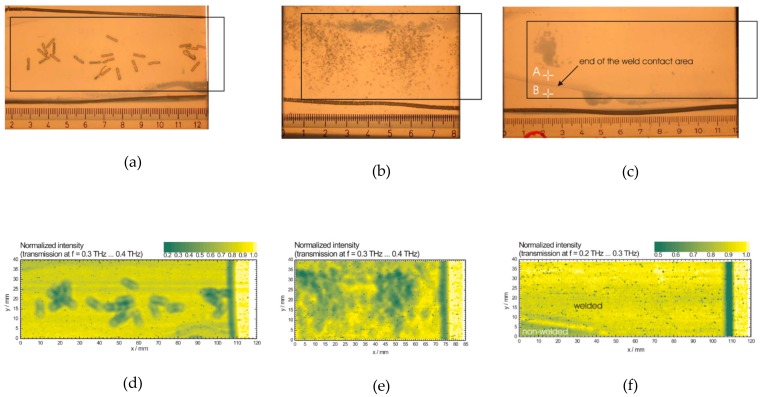
THz TDS transmission imaging of foreign objects between HDPE laminates: (**a**–**c**) Backlighted photograph of metal staples, sand, and air between welded HDPE laminates, respectively; (**d**–**f**) TDS imaging of the same sample above. Foreign objects and air can be seen as areas with low transmitted intensity [48].

**Figure 5 sensors-20-00712-f005:**
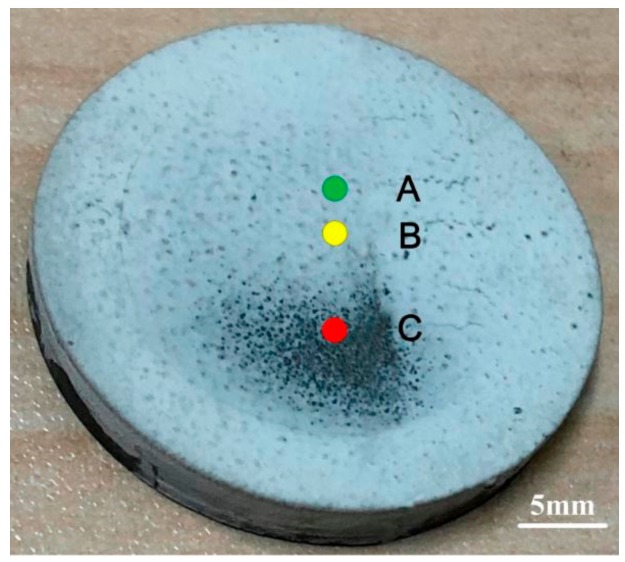
Photograph of the thermal barrier coating sample after artificial erosion. Spots A, B, and C has undergone increasing degrees of erosion and were scanned using THz reflection spectroscopy [50].

**Figure 6 sensors-20-00712-f006:**
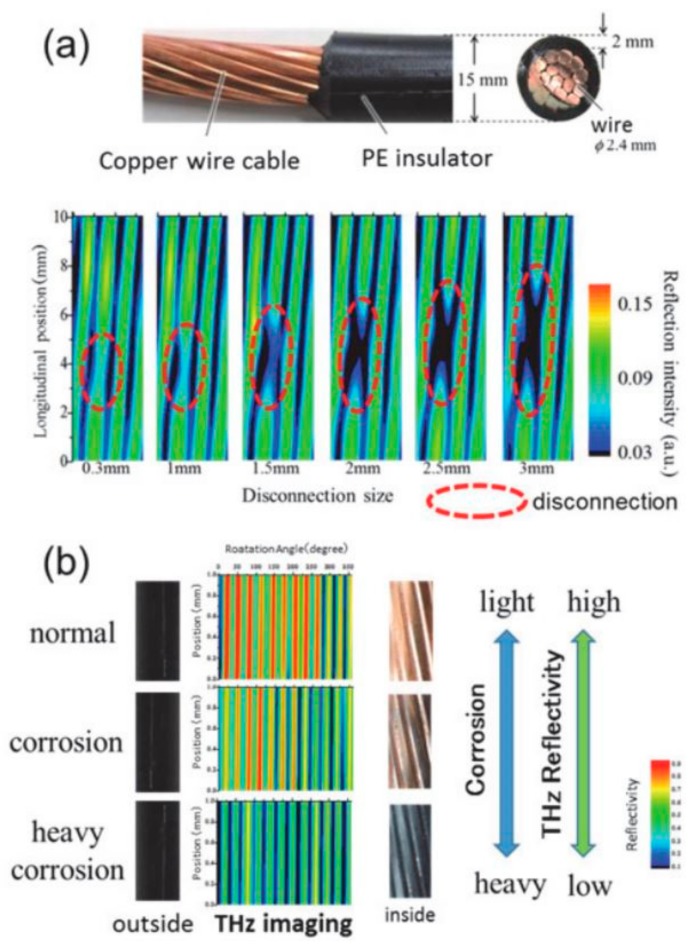
(**a**) Photograph of the insulated copper cable (inset) and THz reflection images of insulated cables with varying disconnection sizes; (**b**) THz reflection images of insulated cables with varying degrees of corrosion. Reproduced with permission from [41].

**Figure 7 sensors-20-00712-f007:**
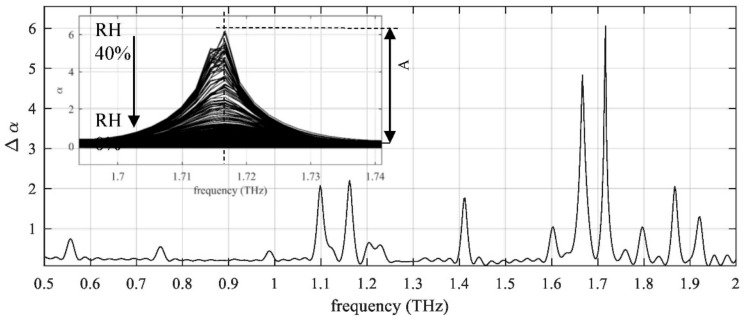
The relative absorption measured at an ambient relative humidity (RH) at 1 atm, 275 K. The inset shows the change of the absorption line at 1.716 THz when the RH decreases from 40% to 0% [54].

**Figure 8 sensors-20-00712-f008:**
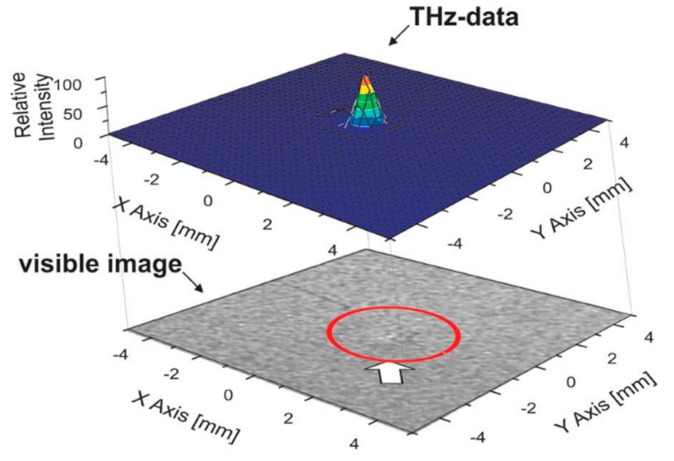
THz dark-field image (top) of a small protrusion on the surface of a steel sheet sample, and the visible image of the same sample (bottom). In the visible image, the protrusion is barely visible (circled in red), whereas it is visible in the THz dark-field image. Reproduced with permission from [56].

**Figure 9 sensors-20-00712-f009:**
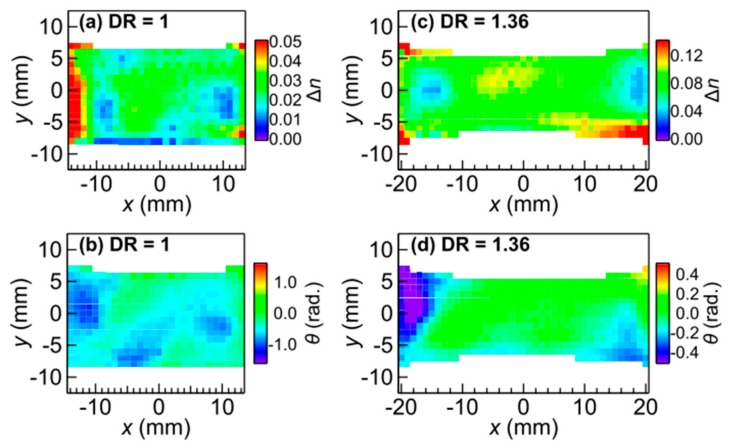
Spatial map of the change in refractive index (Δn) and the slow optical axis (θ) with respect to the *x*-axis for a black rubber sample under normal condition versus stretched to 1.36 times (DR = 1.36) its length in the *x*-axis: (**a**,**b**) Δn is mostly around 0.03, and θ is mostly random; (**c**,**d**) Δn peaks in the centre where it has been stretched the most, and θ has become uniformly zero in the centre. Reproduced with permission from [62].

**Table 1 sensors-20-00712-t001:** Loss of thickness (∆d) calculated from time delay (∆T) spot positions A, B, C compared to digital micrometer measurements [50].

Experimental Results	Intact	A	B	C
First reflection peal time (ps)	13.71 ± 0.11	14.13	14.35	15.20
Time interval ∆T (ps)	-	0.43	0.64	1.49
Loss of thickness ∆d (µm)	-	63.99	96.00	224.00
Electronic digital readout micrometer (µm)	-	70	106	228

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
