# Peer review of "Non-Contact, Non-Destructive Testing in Various Industrial Sectors with Terahertz Technology"

_sensors, 2020, doi:10.3390/s20030712_

Round 1
Reviewer 1 Report
The manuscript gives general introduction to the typical application of THz imaging and spectroscopy techniques in three typical heavy industries, the building and construction industry, the energy and power industry, and the manufacturing industry. The implemented THz imaging and spectroscopy techniques mainly include CW wave imaging in either transmission mode or back-reflection mode, spectral resolved imaging by TDS system. Material characterization, ranging from embedded object imaging in concrete, defect detecting in foam, HDPE plate, cable, measurement of coating thickness, gas concentration, and identification of metal plate surface protrusion, polymer composites, etc. At the end, the authors provide some prospective view about the future trends of the development of THz technology. The manuscript is valuable for the populating the knowledge of the basic concept of THz technology and its application. Hence I would like to recommend it to be published in Sensors. However, some major revisions are strongly requested.
Most important, as for a review paper, the manuscript in the current form is too rough. There is a lack of some important information about the application of THz techniques. For example: (1)key parameters, such as imaging or spectral resolution; (2)limitation of these parameters; (3)if the current performance has met the meet the actual requirement of the applications in industry; (4)potential methods to improve these parameters to meet the actual applications in industry. Same for the session of future outlook. Since the authors has focused on the application of THz technology in heavy industries, major obstacles for wide application of THz techniques should be emphasized at the beginning of this session. Which is mainly considered, pricing issue or technique performance issue? Then the outlook of the future development of THz technology should be closely related to the requirements in the building and construction industry, the energy and power industry, and the manufacturing industry. Some minor suggestions: 10^12 frequencies should be frequencies around 10^12 Hz The general definition of THz wave covers parts of millimeter wave and sub millimeter wave. However, THz wave has never been referred to as millimeter wave or sub millimeter wave. The authors mentioned that “thermal noise, which is typically 6.3 THz and at 6.5 times the energy of a THz photon at room temperature”. However, it is confusion. The authors may need to clarify this point. Particularly, why is it not 6.3 times the photon energy of 1 THz? Absorption of both water and chloride ions are considered of the characterization of concrete. How would they influence the performance of the imaging, such as depth, contrast? There is severe lack of references. It is strongly suggested to cite a few representatively references for each part of the introduction. For example, when mentioning some new materials are developing for THz cameras, a few examples and their key performance should be clearly mentionedAuthor Response
Please see the attachment.
We thank you for your comments and your time.
Best regards
Yu Heng Tao

Reviewer 2 Report
The paper from Tao et al. contains a review on the NDT methods of potential interest for heavy industrial applications and based on THz technology.
Authors makes quite a comprehensive job even if they survey only selected fields: Building and Construction, Energy and Power, Manufacturing.
In this respect, however, they spent a very limited time discussing the potential of QCL sources for THz-CW systems. QCL sources, even if still not mature, may open interesting perspectives for the developments of compact, inexpensive, easy to put in production line systems. I suggest to give some emphasis on the pros and cons of this technology. A nice review on THz QCL is the paper by Rauter and Capasso, “Multi-wavelength quantum cascade laser arrays,” Laser Photonics Rev. 9, 452 (2015).
An additional topic authors might want to consider - adding a paragraph - is the use of novel imaging techniques based on compressive sampling, for example for crack detection (Angrisani et al., “First steps towards an innovative compressive sampling based-THz imaging system for early crack detection on aerospace plates”, IEEE Metrology for Aerospace, 2014) or on deconvolution methods for quality control (Ahi et al., “Quality control and authentication of packaged integrated circuits using enhanced-spatial-resolution terahertz time-domain spectroscopy and imaging,”).
Finally, one more suggestion is to mention the use of THz technology in automotive industry (as part of Manufacturing), as for example in Krimi et al., “Highly accurate thickness measurement of multi-layered automotive paints using terahertz technology,” Appl. Phys. Lett. 109, 021105 (2016).
Miscellanea:
- in lines 64-65, authors state that the output power for CW systems is one order of magnitude larger than in TDS. However, power in the latter systems rarely exceed some tens of microWatt, so that difference is by far larger
- please specify the acronym RH the first time you mention relative humidity (line 284)
- in line 292, please specify the missing ref
- there are some misprints that need to be corrected in lines: 73 (6.5 times), 137 (applications), 297 (In Figure 8), 315 (sentence unclear), 367 (using THz cameras)
Once authors have considered these minor revisions, the manuscript can be published.
Author Response
Please see the attachment.
We thank you for your comments and your time.
Best regards
Yu Heng Tao

Reviewer 3 Report
The authors of this review article describe a small selection of interesting applications of terahertz technologies in the context of heavy industry. Unfortunately, the manuscript has many shortcomings, especially with regard to the industrial focus, and needs to be completely revised.
The introduction to the terahertz spectrum is unfortunately very generic and does not reflect the essential technological advances that make a broad industrial application possible. Although the special importance of the introduction of femtosecond lasers (which took place more than 30 years ago!) is referred to, only the availability of fiber-coupled laser systems (pulsed and continuous wave) and the increasing availability of electronic high-frequency systems as well as MMICs offer attractive possibilities for the realization of industrial system solutions. Today, such systems dominate the terahertz market and are used in the field of non-destructive terahertz testing. These aspects must be sufficiently addressed in a review with a clear reference to heavy industry.
The following distinction between pulsed or TDS and CW systems makes sense due to the different metrological procedures, but is partly misrepresented. While pulsed terahertz systems are naturally very broadband, CW systems enable frequency-selective measurements and can also achieve high bandwidths > 2 THz when using tunable two-color lasers, for example. On the other hand, electronic CW systems are limited in their bandwidth, but are characterized, for example, by better integration capability (MMICs) and higher frequency resolution in coherent measurement setups. In contrast to the manuscript, the use of phase information is also widespread in CW systems (e.g. for thickness measurements of tubes) and therefore does not constitute a criterion for differentiation from TDS systems. In addition, CW terahertz spectroscopy systems have also successfully established themselves on the market. The relation to the higher output power of CW systems is strongly dependent on the used terahertz frequency and the underlying technology. Furthermore, the functionality of a TDS system is roughly outlined, but only different CW technologies are dealt with arbitrarily. In the NDT area, various electronic systems as well as fiber-coupled two-color laser and TDS systems have established themselves. A more differentiated presentation with regard to the underlying topic is absolutely necessary.
In the manuscript, many formulations are too casual and the content is not sufficiently differentiated or simply incorrect. Especially in the field of non-destructive testing a multitude of different methods are used. A distinction must be made between the thermal, acoustic and electrical conductivity of materials, or at least reference must be made directly to the latter (line 32). Figure 2 shows the optical properties of a foam insulation, which is not described in more detail and has been generalized according to the textual expression. However, a concrete designation is absolutely necessary, since apart from the type of foam, the material plays a decisive role. There are many examples in the literature which show a much stronger absorption behaviour than the illustrated type of insulation foam. If necessary, reference can also be made here to the refractive index curve shown. In lines 185-186 it is described that today there are routine checks of the thermal insulation of space shuttles before launch, but the space shuttles were discontinued years ago! Unfortunately, there are numerous other inconsistencies in the manuscript, which can often be corrected by a more skilful use of language. For example, contrary to the text in lines 369-371, real-time terahertz cameras cannot only be implemented thermally or FET-based! The complete section on the future outlook again seems very arbitrary. In line 373, EO crystals are mentioned in connection with CCD cameras as if they were part of CCDs. The concept existed over 10 years ago, but does not reflect the state of the art. Also here, as in all previous sections, a stronger differentiation has to be made with regard to the application topic.
The manuscript needs a complete revision of content and language before it can be recommended for publication.
Author Response
Please see the attachment.
We sincerely thank you for your comments and your time.
Best regards
Yu Heng Tao

Reviewer 4 Report
The authors conduct a very impressive and wide-ranging demonstration of THz imaging technology, and carefully consider possibilities for its integration into a commercialized THz system. The manuscript contains a few very minor English-language errors (which can easily be taken care of by the journal's pre-publication subediting process). Otherwise, the manuscript is ready for publication in Sensors as-is.
Author Response
We thank you for your compliments and your time.
Best regards
Yu Heng Tao
Round 2
Reviewer 1 Report
My previous questions and comments have been properly considered.
Reviewer 3 Report
My previous comments have been properly considered.